# Loss of *atm* in Zebrafish as a Model of Ataxia–Telangiectasia Syndrome

**DOI:** 10.3390/biomedicines10020392

**Published:** 2022-02-03

**Authors:** Kehua Chen, Peng Wang, Jingrun Chen, Yiling Ying, Yi Chen, Eric Gilson, Yiming Lu, Jing Ye

**Affiliations:** 1Department of Geriatrics, Shanghai Ruijin Hospital, Shanghai Jiaotong University School of Medicine, Shanghai 200001, China; chenkehua@sjtu.edu.cn (K.C.); wangp123@sjtu.edu.cn (P.W.); yusaki@sjtu.edu.cn (J.C.); yylb0985@rjh.com.cn (Y.Y.); 2Medical Center on Aging of Ruijin Hospital, Shanghai Ruijin Hospital, Shanghai Jiaotong University School of Medicine, Shanghai 200001, China; 3International Laboratory in Hematology and Cancer, School of Medicine, Shanghai Jiao Tong University, Shanghai 200001, China; chenyi1827@sina.com (Y.C.); eric.gilson@unice.fr (E.G.); 4Faculty of Medicine, University Côte d’Azur, CNRS, INSERM, IRCAN, 06107 Nice, France; 5Department of Medical Genetics, CHU, 06107 Nice, France

**Keywords:** *ATM*, regeneration, tumorigenesis, immune deficiency, zebrafish

## Abstract

Ataxia–telangiectasia mutated (*ATM*) is a key DNA damage signaling kinase that is mutated in humans with ataxia–telangiectasia (A-T) syndrome. This syndrome is characterized by neurodegeneration, immune abnormality, cancer predisposition, and premature aging. To better understand the function of *ATM* in vivo, we engineered a viable zebrafish model with a mutated *atm* gene. Zebrafish *atm* loss-of-function mutants show characteristic features of A-T-like motor disturbance, including coordination disorders, immunodeficiency, and tumorigenesis. The immunological disorder of *atm* homozygote fish is linked to the developmental blockade of hematopoiesis, which occurs at the adulthood stage and results in a decrease in infection defense but, with little effect on wound healing. Malignant neoplasms found in *atm* mutant fish were mainly nerve sheath tumors and myeloid leukemia, which rarely occur in A-T patients or *Atm−/−* mice. These results underscore the importance of *atm* during immune cell development. This zebrafish A-T model opens up a pathway to an improved understanding of the molecular basis of tumorigenesis in A-T and the cellular role of *atm*.

## 1. Introduction

Ataxia–telangiectasia (A-T) is a rare autosomal recessive disorder with a complex phenotype, including neural degeneration accompanied by ataxia (movement dysfunction), immunodeficiency, thymic and gonadal atrophy, radiation sensitivity, and predisposition to cancer [1]. A-T is caused by nonsense or missense mutations in the ataxia–telangiectasia-mutated (*ATM*) gene, which encodes a serine/threonine protein kinase [2] with critical roles in DNA double-strand break (DSB) repair, genomic stability, cell cycle regulation, reactive oxygen species (ROS) regulation, and cell survival [3]. Mice, rats, and flies with mutations of *ATM* have shown A-T characteristics, including neural degeneration, meiotic defects, radiation hypersensitivity, immunological abnormalities, and cancer [4,5,6,7]. These data highlight the key roles of *ATM* in neurodevelopment, immunity, and reproduction.

Immunological dysfunction and tumorigenesis have been well characterized in A-T patients and the *Atm* mutation mouse model [4,8]. Immunological deficiency in A-T patients is highly variable, but the majority of cases show T cell lymphopenia as well as immunoglobulin disorders, such as IgG or IgA deficiency or high titers of IgM [9,10]. Such pathological variation disturbs the homeostasis of both T and B cells, resulting in reduced immune repertoire diversity and suboptimal T and B cell interactions [9,11], in turn causing infection and autoimmune disease [12,13]. A study in *Atm* mutant mice also showed a decreasing number of lymphocytes, indicating a lymphoid cell development disorder [5,14,15]. The immunodeficiency of A-T patients results in a predisposition to tumors, which develop in 10–20% of cases [16,17]. The most prevalent types of malignancy in A-T patients are leukemia and lymphoma, most notably acute lymphoblastic leukemias (ALLs) [18]. Although adult A-T patients may exhibit solid tumors including breast, gastric, or esophageal carcinoma, such cases are rare [1]. In a mouse model, T cell lymphoblastic leukemias and lymphoma were observed in several *ATM* mutations; no other types of tumors were reported [5,19]. Although many studies have addressed the immunological function of *ATM*, little is known about its role in immune cell development in early embryonic stages or during oncogenesis.

Zebrafish is recognized as a model for genetic studies in the fields of developmental biology, neurodegeneration, and cancer [20,21,22]. A previous study showed that the zebrafish *ATM* protein is highly conserved within vertebrates. In vivo knockdown of *atm* in zebrafish embryos using antisense morpholino oligonucleotides resulted in developmental abnormalities and irradiation sensitivity [23]. To fully address *atm* function in zebrafish and its role in the immune system, we established an *atm*-mutated zebrafish model using the CRISPR–Cas9 system. *atm* homozygote zebrafish (*atm*−/−) were all male and infertile. Loss of *atm* also affected motor function and coordination of zebrafish, similar to other species [1,4]. Furthermore, we showed that the mutation of *atm* can cause a global disorder of immune cell development, i.e., not just in lymphocytes, and impair the immune defense in adulthood. Notably, *atm−/−* fish have seemingly normal regenerative capacity, although cytokine induction upon injury is attenuated but not abolished. This is consistent with the fact that zebrafish is a highly regenerative organism [24] capable of regeneration even in the context of an attenuated inflammation response [25,26,27,28]. Finally, we show that *atm* mutation in zebrafish results in a different spectrum of tumors compared to mouse models, such as malignant peripheral nerve sheath tumors and myeloid leukemia. Such malignancies also occur in A-T patients [29,30,31,32], revealing that the zebrafish, due to its similarities to humans, is a more efficient model than mice in studying the tumor formation caused by *atm* mutation, thus providing a novel and interesting *ATM* vertebrate model of malignancies.

## 2. Materials and Methods

### 2.1. Zebrafish Maintenance and the Generation of Mutants

Tubingen wild-type zebrafish strain was acquired from Shanghai Institute of Hematology. Zebrafish were fed a regular diet and raised and maintained under standard conditions as described previously [33]. Animal procedures were approved and performed according to the Animal Care and Use Committee of Shanghai Jiao Tong University. The protocol was also approved by the Ethics Committee of Ruijin Hospital affiliated with the School of Medicine at Shanghai Jiao Tong University, China. ARRIVE 2.0 Essential 10-item set is provided in the Appendix A. *Atm* mutant zebrafish were generated using the CRISPR–Cas9 system, in which a guide RNA targeting exon 17 of *atm* (sgRNA: 5′-TGCGTCTTCGGAGCTCAACGG-3′) was designed using ZiFiT Targeter software (http://zifit.partners.org/ZiFiT, 5 August 2017). The guide RNA was synthesized by cloning annealed oligonucleotides into the sgRNA vector as described previously [34]. Genotyping was performed on the genomic DNA isolated from the tails of zebrafish to identify *atm* mutants (primers for genotyping: fwd 5′-CAAACGAGCAGAATCCAA-3′; rev 5′-TACAGATGCCACCGAAAT-3′).

### 2.2. Hematoxylin–Eosin Staining

Zebrafish tissues were isolated and fixed in paraformaldehyde for about 24–48 h and washed repeatedly in 70% ethanol. Tissues were dehydrated and embedded in paraffin and sectioned at 5 microns. Slices were baked at 70 °C for 4 h and deparaffinized in dimethylbenzene 3 × 15 min. Hematoxylin–eosin staining (C0105S, Beyotime, Shanghai, China) was performed according to the manufacturer’s instructions. Then, the slices were stained with hematoxylin for 10 min and washed with water for 2 × 10 min. Next, eosin was applied for 1 min. At last, slices went through ethanol gradual dehydration and dimethylbenzene treatment 2 × 10 min. Then, neutral gum was used to mount slides. Images were taken with Zeiss A2 microscope (Zeiss, Oberkochen, Germany).

### 2.3. Anti-Acid Staining

Zebrafish tissues were separated then smeared on the slices. Kinyoun cold dyeing method was used to detect Acid fast bacilli. In brief, slices were stained in Kinyoun dye solution for 5 min and washed with distilled water. Slices were incubated in Kinyoun decolorizing solution for 5 min and washed with distilled water. Finally, slices were stained with methylene blue followed by washing with distilled water. Images were taken with Zeiss A2 microscope (Zeiss, Oberkochen, Germany).

### 2.4. Zebrafish Behavioral Assays

An adult fish was placed in a standard mating tank (21 × 10 × 7.5 cm) containing system water to a depth of 6 cm and allowed to acclimatize for 15 min and then transferred to an automated observation and video-tracking system (ZebraLab; ViewPoint Life Sciences, Lyon, France) for an open-field tracking test. The movement of the zebrafish was monitored continuously for 1 h. The average speed was recorded and analyzed using video-tracking software (ViewPoint Life Sciences, Lyon, France). The activity of the zebrafish was monitored continuously for 1 h. Quantization tests focused on the activity of zebrafish, recording the total amount of movement of zebrafish in the tanks and its frequency. The software can automatically record the position of the zebrafish and compare the previous position with the new position. The changed pixels were regarded as the activity. We used the wild-type fish, obtaining two thresholds’ values (burst threshold value was 100 and freezing value was 20). The burst threshold value indicates that the pixel change of 95% movement is below this value (black dotted line). If the value of the moving surface was above the burst threshold, the activity was recorded as highly active, and the software automatically recorded the duration of the activity. (ViewPoint Life Sciences, Lyon, France).

### 2.5. Quantitative RT-PCR

Total RNA was extracted from caudal fins using a TRIZOL reagent. cDNA synthesis was carried out using an RT SuperMix kit (R323-01, Vazyme, Nanjing, China) according to the manufacturer’s instructions. Quantitative RT-PCR was performed using SYBR Green (Q711-02/03, Vazyme, Nanjing, China). Expression profile of genes including *atm*, *β-actin*, *il-1β*, *il-4*, *il-6*, *il-8*, *il-10*, *il-13*, *tnf-α*, *ifn-γ*, *tgf-1β*, *mmp9*, *mmp13a*, *plaua*, *plaub*, *ctsd gpx1b*, *sod1*, and *sod2* was analyzed using the primers listed in the table below.
*atm*-F (PCR1): 5′-GAGTTGGCCCTCTGTCATGC-3′*atm*-R (PCR1): 5′-TATCAACGGGCACCAGGGAT-3′*atm*-F (PCR2): 5′-TGATTCTTTGCCCTTGGCCC-3′*atm*-R (PCR2): 5′-AACTGGTCCAAGTTCCCCCA-3′*atm*-F (PCR3): 5′-TCACTTCCAGACCTTCCCAGT-3′*atm*-R (PCR3): 5′-AGGAGAATCCACCGTTGAGCT-3′*il-1**β*-F: 5′-TCCGCTCCACATCTCGTACT-3′*il-1**β*-R: 5′-AACCGGGACATTTGACGGAC-3′*il6*-F: 5′-AGCAGGAATGGCTTTGAAGGG-3′*il6*-R: 5′-GTCAGGACGCTGTAGATTCGC-3′*il8*-F: 5′-GTAGATCCACGCTGTCGCTG-3′*il8*-R: 5′-TACAGTGTGGGCTTGGAGGG-3′*tnfa*-F: 5′-AGACCTTAGACTGGAGAGATGAC-3′*tnfa*-R: 5′-CAAAGACACCTGGCTGTAGAC-3′*il10*-F: 5′-GGAGACCATTCTGCCAACAGC-3′*il10*-R: 5′-TCTTGCATTTCACCATATCCCG-3′*β-actin*-F: 5′-GGGTATGGAATCTTGCGGTATC-3′*β-actin*-R: 5′-CTTCATGGTGGAAGGAGCAA-3′*tgf1β*-F: 5′-GCTGGAACTGTATCGCGGAG-3′*tgf1β*-R: 5′-ATCCGTGCTCTGCTGGTTTG-3′*il4*-F: 5′-AGCAGGAATGGCTTTGAAGGG-3′*il4*-R: 5′-TTCATTGTGCATTCCCCCGAG-3′*il13*-F: 5′-GCACTGTATTCGTCTCGGGT-3′*il13*-R: 5′-CGGACAGGCCAGAAACCTTTT-3′*mmp9*-F: 5′-GAGGGCCGCAATGATGGAAA-3′*mmp9*-R: 5′-CGATGCAAGGGGAAACCCTC-3′*mmp13a*-F: 5′-GCTGCGGAGTTCCAGACATC-3′*mmp13a*-R: 5′-AGTAACGTCAGCCCACACCT-3′*plaua*-F: 5′-CTCAGCCATGGTGCGTTGTA-3′*plaua*-R: 5′-TTTGTCTGTCGGTCCAAGCG-3′*plaub*-F: 5′-ACCGAGAACATGCTGTGTGC-3′*plaub*-R: 5′-GGCATAAACACCAGGCCGAA-3′*ctsd*-F: 5′-CGGGTATCTCAGCCAGGACA-3′*ctsd*-R: 5′-AGGAGGAACCCCATCGACAG-3′*gpx1b*-F: 5′-CAGATGAACGAGCTGCACGA-3′*gpx1b*-R: 5′-GGGCTCATAGCCATTGCCAG-3′*sod1*-F: 5′-ACCTGGGTAATGTGACCGCT-3′*sod1*-R: 5′-TCATTGCCACCCTTCCCCAA-3′*sod2*-F: 5′-AACCCCCTGTTAGGTGCTGT-3′*sod2*-R: 5′-GTTGCATGGTGCTTGCTGTG-3′

### 2.6. RNAscope^®^ In Situ Hybridization Combined with Immunofluorescence

RNA Protein Co-detection Ancillary Kit (323180, Advanced Cell Diagnostics, Newark, CA, USA) and RNAscope^®^ Multiplex Fluorescent Reagent Kit v2 (323100, Advanced Cell Diagnostics, Newark, CA, USA) were used in accordance with the manufacturer’s instructions [35]. In brief, tissue was separated and embedded in OCT. Tissue was sectioned at 6 μm. Slices were fixed in 4% PFA at 4 °C for 15 min. Sections were dehydrated, then incubated with RNAscope^®^ Hydrogen Peroxide for 10 min at room temperature, and then incubated with primary antibody GFAP (GTX128741, GeneTex, Alton Pkwy, Irvine, CA, USA) at 4 °C overnight. After three washes in PBS-T, sections were incubated with RNAscope^®^ Protease Ⅳ at 40 °C for 30 min in a HybEZ hybridization oven (Advanced Cell Diagnostics, Newark, CA, USA). Hybridization with target probes (DR-tnfa Cat. 575111. Advanced Cell Diagnostics, Newark, CA, USA) was carried out incubating the slides at 40 °C for 2 h, and then the slides were incubated at 40 °C with the following reagents: Amplifier 1 (30 min), Amplifier 2 (30 min), Amplifier 3 (15 min); HRP-C1 (15 min), TSA^®^ Plus fluorophore for channel 1 (fluorescein, cyanine 3, or cyanine 5, PerkinElmer, Waltham, UK; 1:1000; 30 min), and HRP blocker (15 min). After each hybridization step, slides were washed twice with Wash Buffer at room temperature. After hybridization, slices were incubated with fluorescently labeled second antibody at room temperature for 1 h and then stained with DAPI. Slices were mounted in VECTASHIELD and examined under Zeiss A2 microscope.

### 2.7. Whole-Mount In Situ Hybridization

Whole-mount mRNA in situ hybridization was conducted as described previously [36]. Digoxigenin (DIG)-labeled RNA probes against *gata1*, *mpo*, and *rag1* were transcribed with T7 or T3 polymerase (11031163001 or 10881767001, Roche, Basle, Switzerland). Probes labeled with DIG (11277073910, Roche, Basel, Switzerland) were detected using anti-digoxigenin Fab fragment antibody (Roche) with 5-bromo-4-chloro-3-indolyl phosphate–nitroblue tetrazolium (BCIP/NBT) staining (SK-5400, Vector Laboratories, Burlingame, CA, USA).

### 2.8. Senescence-Associated β-Galactosidase Activity

Determination of SA-β-gal activity was performed through the widely used X-gal staining method and by a chemiluminescent reaction employing Galacton substrate (C0602, Beyotime, Shanghai, China). For the chromogenic reaction, adult caudal fins were amputated and fixed in β-galactosidase fixative overnight. After fixation, caudal fins were washed 3 times in PBS (10 min per time) and stained with freshly prepared β-galactosidase staining working solution as indicated by the kit protocol and incubated at 37 °C for 6 h before observation.

### 2.9. Flow Cytometry Analysis

Kidney cells isolated from zebrafish were resuspended in 0.9× phosphate-buffered saline (PBS) + 5% fetal bovine serum (FBS) and passed through a 40 µm filter. Cells were washed in ice-cold 0.9× PBS + 5% FBS and flow cytometry was performed on BD FACSCalibur. FSC voltage was set to 500, linear display. SSC voltage was set to 200, logarithmic display.

### 2.10. Immunohistochemistry

Paraffin sections (5 µm) were used for immunohistochemical determination of mpx expression. Slides were baked at 70 °C for 4 h and deparaffinized in dimethylbenzene 3 × 15 min. Then, slides were hydrated in ethanol, and immunohistochemistry was performed according to the manufacturer’s instructions (D601037, Sangon Biotech, Shanghai, China). Slides were treated with sodium citrate buffer for a heat-induced epitope retrieval step in a pressure cooker, boiling for 20 min. After antigen retrieval, slides were washed in PBST for 5 min and pretreated with peroxidase suppressor reagent to quench endogenous peroxidase activity, followed by 3%BSA in PBST blocking for 30 min. Blocking reagent discarded, primary antibody against zebrafish mpx (GeneTex, GTX128379, Alton Pkwy, Irvine, CA, USA) was applied at a 1:400 dilution overnight. Slides were rinsed 3 × 5 min in PBST. Goat antibody was applied to rabbit horseradish peroxidase-conjugated antibody for 60 min, after which slides were washed 3 × 5 min in PBST. Immuno-peroxidase staining was developed with diaminobenzidine (DAB)+ chromogen according to the manufacturer’s instructions. After hematoxylin counterstaining for 5 min and 1% hydrochloric acid–ethanol for 3~5 s for depigmentation, slides were then treated in PBST for reverse blue staining. After ethanol gradual dehydration and dimethylbenzene treatment 2 × 10 min, neutral gum was used to mount slides. Images were taken with Zeiss A2 microscope (Zeiss, Oberkochen, Germany).

### 2.11. Quantification and Statistical Analysis Quantification and Statistical Analysis

The data were analyzed by GraphPad Prism software. Data used in this study passed normality and lognormality tests. The statistical significance between two groups was determined using unpaired Student’s t-test, with tow-tailed *p* value. Among three or more groups, one-way analysis of variance followed by Bonferroni’s multiple comparison test or Dunnett’s multiple comparison test was used for comparisons.

## 3. Results

### 3.1. Establishment of an atm Fish Model

To study the immunological function and tumorigenic implications of *atm* in zebrafish, we mutated *atm* using CRISPR–Cas9. Guide RNAs were designed to target the exon 17 sequence, resulting in an insertion of 11 bp (Figure 1A). This mutation led to a premature stop codon, resulting in truncation of the protein and loss of its functional domain (Figure 1A). We designed three pairs of primers to target the 5′ terminus (PCR1), 3′ terminus (PCR2), and guide RNA (PCR3) regions on the *atm* transcript to assay knockout efficiency. RT-qPCR showed that *atm* expression was decreased in *atm* heterozygote fish (*atm*+/−) and further downregulated in *atm* homozygote fish (*atm*−/−) (Figure 1B). This is likely because the mutated transcript rapidly leads to RNA instability and degradation. PCR3 was undetected in *atm*−/−, demonstrating full knockout of the functional protein (Figure 1B).

Adult *atm*−/− fish (6 months old) did not show any morphological differences but underwent sex reversal and were all male (Figure 1C,D). Such a biased sex ratio is a peculiar characteristic of DDR-compromised zebrafish [37]. These fish could not fertilize wild-type (WT) oocytes, while those that were outcrossed indicated the infertility of *ATM*−/− fish (Appendix A). Indeed, dissection showed that the size of the testes of *atm*−/− fish was reduced compared to WT (Appendix A). Hematoxylin and eosin staining indicated that *atm*−/− fish lacked mature spermatozoa and showed meiosis blockage in the spermatocyte state (Figure 1E,F). This infertility phenotype can be explained by the role of *Atm* in chromatid association during meiosis [5].

### 3.2. ATM−/− Fish Show Motor Disability and Coordination Disorder

Loss of the *ATM* gene in humans causes ataxia at an early age [38]; *Atm* mutant mice also show coordination disability [4]. These data indicate a crucial role of *ATM* in brain development and function. To address whether the knockout of *atm* affects fish behavior, we performed a 60-minute open field test on adult (6-month-old) WT, *atm*+/−, and *atm*−/− fish. Motion tail diagrams showed that the exploration ability of *atm*+/- and *atm*−/− fish was intact, since the tails were evenly distributed; the prevalence of low-speed tails was higher in homozygote fish (Figure 2A). The average speed of *atm*−/− fish was decreased compared to WT (Figure 2B), indicating that motor function was impaired after *atm* knockout. We then tested the activity levels of these fish (see Methods). Interestingly, *atm*−/− fish showed lower activity levels during the 60-minute test (Figure 2C), and their highly active period was shorter than that of WT fish (Figure 2D). Such behavior reflects a coordination disorder, since fish tend to be rigid while swimming. These data suggest that loss of *atm* in zebrafish can cause similar behavioral disabilities, such as motor and coordination disorders, in humans and mice, and further indicate an important role of *atm* in neural function.

Previous studies have shown that oxidative stress and neural inflammatory in the brain are key pathologies of A-T brain dysfunction [6,7,39]. In our zebrafish A-T model, we found that cytokines such as *tnfa*, *il-6*, and *il-8* were increased in the cerebellum and hindbrain of *atm*−/− fish (Appendix A). We revealed that the hindbrain was enriched in GFAP-positive glial cells with a transcriptional elevated level of *tnfa* at the single-cell level with the RNA-scope (Appendix A). This suggests that glial cell inflammatory symptoms in A-T progress [7]. Moreover, the expression of genes related to redox homeostasis such as *gpx1b*, *sod1*, and *sod2* was elevated in mutant fish (Appendix A), suggesting oxidative stress participated in the neural pathophysiology of A-T. These data further prove *atm* is an essential gene regulating redox and inflammatory hemostasis. Loss of *atm* in the brain can cause oxidative stress and neural inflammation, resulting in neurological phenotype of A-T.

### 3.3. Lack of atm Causes Immune Deficiency during Zebrafish Development

We observed a shortened life span in *atm*−/− fish; premature death started in the juvenile stage (Figure 3A). Most *atm*−/− fish died of a systemic infection before reaching 12 months of age or had a malignant tumor at the mid-age stage (Figure 3B), suggesting a fragile immune system. The infected fish all showed abdominal dropsy, swollen kidneys, and anti-acid bacterial colonization in the infected tissue (Figure 3C). Considering that A-T patients have a variety of immunological symptoms, including autoimmune disease [40], consistent inflammation [41], and infection [42,43], and that *Atm* mutant mice show lymphocyte development disorder [14], *atm*−/− zebrafish may have an immune deficiency syndrome similar to other species based on these pathological changes.

To address the immune development of *atm*−/− fish, we performed whole-mount in situ hybridization (WISH) analyses on fish embryos 24 h post-fertilization (hpf) to 5 days post-fertilization (dpf), using the probes *gata1* (erythroblast marker), *mpo* (mature neutrophil marker), and *rag1* (lymphocyte marker). We observed a *gata1*-positive erythroblast cluster in caudal hematopoietic tissue at 24-hpf; thus, loss of *atm* did not affect erythroblast development (Figure 3D). However, *atm*−/− embryos showed less *mpo*-positive cells at 24-hpf and decreased *rag1* intensity in the thymus at 5-dpf (Figure 3E,F) compared to WT. These findings indicate that in *atm*−/−, myeloid and lymphoid cell lineages are impaired at the embryo stage, but erythroid cell lineage appear to remain intact.

### 3.4. Lack of atm Does Not Impair Fish Tail Regeneration

To explore the regenerative capacities of *atm*−/− fish, we amputated fish tails and assessed wound regeneration ability. As wound healing is a complex process that requires several inflammatory and immune cells, including neutrophils and macrophages, excessive or suppressed inflammation could delay healing, affect pigmentation, and disturb tissue repair [44,45,46]. However, the *atm*−/− fish showed similar recovery ability to WT (Figure 3G) at 10 days post-amputation. Nevertheless, pigmentation was delayed in *atm*−/− fish tails (Figure 3G and Table 1). Then, we further addressed the immunological defect in the regenerating tail of *atm*−/− fish by harvesting tissue from WT and *atm*−/− fish 12 h after amputation (Appendix A). Immune cell migration begins within 3 h of wounding [46] and the fibrinolytic system is repressed at this timepoint (Appendix A). Indeed, several cytokines, including interleukin *il-1b*, *il-6*, *il-8*, transforming growth factor *tgfβ-1b*, and matrix metalloproteinases (MMPs), were elevated following amputation of both WT and *atm*−/− fish (Appendix A–E). Nonetheless, inflammation was subdued in *atm*−/− fish compared to WT (Appendix A–E). Cellular senescence also plays an important role in wound repair and tissue regeneration [47,48,49]. Senescent cells exhibit a senescence-associated secretory phenotype, including cytokines that participate in immune cell recruitment, angiogenesis, and epithelialization [50]. To address the level of senescence during tail regeneration in *atm*−/− fish, we measured the SA-β-gal signal at 12 h and 10 days after amputation. However, cellular senescence at the SA-β-gal level was similar between *atm*−/− and WT fish during tail regeneration (Appendix A). Therefore, the regenerative capacities of zebrafish, including the apparition of senescent cells at the injury site, appear in large part *atm*-independent even if the inflammation response is attenuated in *atm*−/− fish.

Overall, our results indicate that *atm*−/− fish showed an immune cell development disorder at the embryo stage, which resulted in immune dysfunction in adulthood. Moreover, our data show that *atm* does not play an essential role in tissue regeneration even if atm loss leads to an attenuated inflammation response to injury.

### 3.5. Atm Loss Promotes Tumorigenesis in Zebrafish

Tumors were the main cause of death in *atm*−/− zebrafish after 12 months (Figure 3B). To investigate neoplasm pathology, we dissected morbid *atm* mutant fish and probed tumor tissue. We found that 100% of the fish had neoplasms in the “trunk” of the kidney (Figure 4A), and hematoxylin and eosin staining of kidney tissue showed enlarged hematopoietic tissue with heterocyst and mitotic figures (Figure 4B). Splenomegaly was also observed in *atm*−/− fish (Figure 4C), with similar heterocyst seen in kidney neoplasms (Figure 4D). These findings suggest that loss of *atm* in zebrafish causes hematopoietic malignancies with metastatic potential in the kidney. By decoding the cellular composition of the kidney tumors of 12-month-old *atm*−/− fish (using flow cytometry), we observed a decrease in the lymphocyte population, as well as an increase in monocytes and precursor cells (Figure 4E). Indeed, the tumor tissue of *atm−/−* fish had a larger *mpx*-positive cell population (Figure 4F) and negative periodic acid–Schiff stain (PAS) (Appendix A), which is considered a sign of myeloid leukemia, compared to the WT kidney. Over-proliferation of monocytes and precursor cells suggests a predisposition to myeloid leukemia in *atm*-mutated zebrafish.

We also observed exophthalmos in 12-month-old and elderly *atm*−/− fish (Figure 4G). Pathological analysis of the neoplasm revealed destruction of the structure, with heterocyst and mitotic figures (Figure 4G). As the majority of the exophthalmos fish exhibited hematopoietic malignancies, the eye neoplasms were likely a result of metastasis. However, the form of the heterocyst in the eye differed from those in the kidney (Figure 4B,G). The eye tumors resembled malignant peripheral nerve sheath tumors (MPNSTs), which are common in p53 and other DNA repair gene fish mutants [51,52,53].

Overall, we found that mutation of *atm* in zebrafish leads to tumorigenesis, thus affecting the survival of fish older than 12 months. Importantly, the malignant neoplasms induced were primarily myeloid hematopoietic malignancies and MPNSTs.

## 4. Discussion

In this study, we engineered an *atm* mutant fish model of A-T syndrome. Loss of *atm* in zebrafish causes meiotic disorders and sex reversal. These phenotypes might be explained by meiotic arrest, DBS accumulation, and oocyte apoptosis [54,55]. Coincidentally, several DNA repair-related genes (e.g., *rad51* [52], *brca2* [56], and *fancI* [57]) have been well characterized in zebrafish; mutations of these genes also result in sex reversal and meiotic defects. These data suggest that *atm* is involved in a pathway similar to other DNA repair genes that play a role in germ cell function. Another representative symptom of A-T is ataxia, which manifests in childhood; increasing difficulty with involuntary movements can occur at any age [58,59]. Evidence has shown that ROS production [60] and glia inflammation [7] may be involved in this pathological process, which results in progressive and diffuse cerebellar degeneration [58,61]. We tested *atm* mutant zebrafish behavior in an open field and found that swim speed and locomotor activity were decreased, which can be considered to reflect motor dysfunction and impaired coordination, thereby emulating the ataxia symptoms of A-T patients.

Our results showed that *atm*−/− fish have an immune cell development disorder. Previous reports mainly focused on lymphoid development in *atm* mutants and suggested that *atm* loss of function leads to lymphopenia and B cell-related immunoglobulin disorder [9,10]. We show here that loss of *atm* contributed to the failure of immune system development in a zebrafish model. Both myeloid and lymphoid hematopoietic cells were reduced at the early embryonic stage. These data suggest that immune deficiency of *atm* loss starts in the embryo.

Although a lack of *atm* reduced inflammation levels in the wound, the capacity of tissue regeneration was maintained. This suggests that the immune cell disorder triggered by *atm* loss may not affect the primary regeneration pathway [62]. Noteworthy, pigmentation was delayed in homozygote fish, indicating some impairment in the global regeneration process. Overall, these results indicate that the regulation of immune function by *atm* only pays a minor role in regeneration.

Tumor predisposition was also evident in *atm−/−* zebrafish. The mutation of *ATM* in humans and mice usually causes ALLs [18]; some adult A-T patients also experience solid tumors including breast, gastric, or esophageal carcinomas [1]. Nonetheless, our *atm* mutant zebrafish primarily suffered from myeloid hematopoietic malignancies and MPNSTs. Although these malignancies have been reported in A-T patients, cases are rare. Neither of these tumors have been observed in the *Atm* mutant mouse model. Thus, the zebrafish, due to its similarities to humans, is a more efficient model than mice in studying the tumor formation caused by *ATM* mutation. Although *ATM* mutation can cause a different spectrum of tumor predisposition in zebrafish, the related mechanisms are not fully understood. As ATM is an upstream kinase regulating DNA damage response, its impairment leads to genomic instability and chromosome rearrangement, which are major risk of oncogenesis. Moreover, ATM can be activated by oxidative stress and antagonize ROS damage [63]. Loss of ATM disturbs balance between the glutathione reduced form and oxidized form by ROS or oxidized thiols, leading to oxidant homeostasis impairment, which contributes to the A-T phenotype, especially cancers [64]. Importantly, the mechanisms by which *atm* mutation can cause a spectrum of tumor predisposition in zebrafish more closely related to A-T patients than transgenic mice remain to be determined. Since ATM connects telomere dysfunction to inflammation and cancer [65], an interesting possibility would be that this difference relies on zebrafish telomere biology, which is more similar to humans than to mice [66].

Our study introduced a viable vertebrate (zebrafish) model of atm loss, which recapitulates many human A-T symptoms. Importantly, *Atm* homozygote zebrafish show mechanistic differences in terms of immunological development and tumorigenesis that are not seen in humans and mice, suggesting a fish-specific mechanism of the *atm*-related pathology. Further study of this mutant will improve our understanding of the cellular roles of *ATM* in vivo and the molecular pathology of A-T.

## Figures and Tables

**Figure 1 biomedicines-10-00392-f001:**
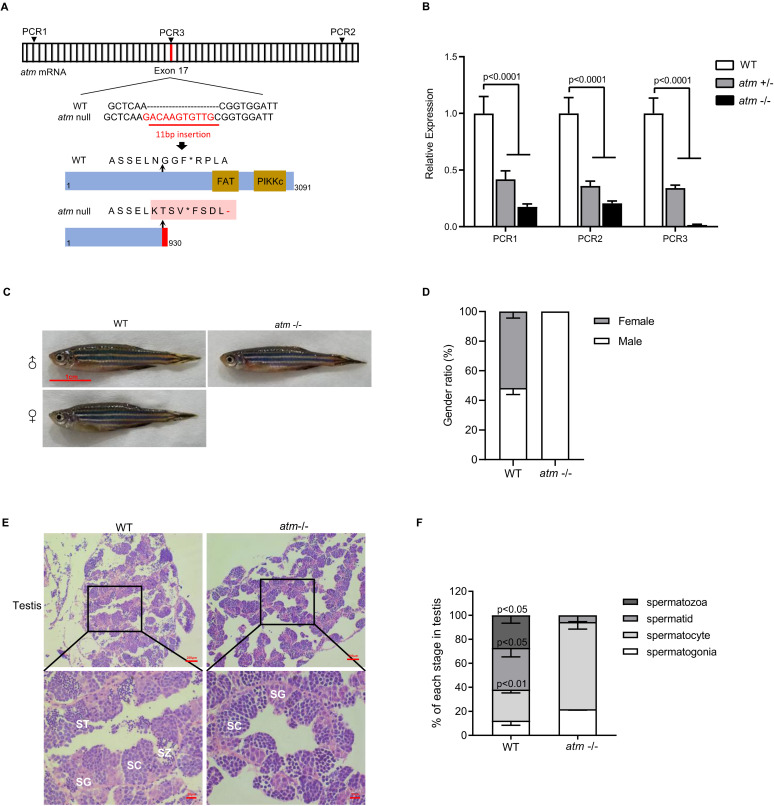
Mutant schema of *atm* and infertility phenotype. (**A**) Cas9-based engineering system of the *atm* introduced a stop code before functional domain. (**B**) Relative expression level of *atm* mRNA at different location (shown in (**A**)) of *atm*+/− and *atm*−/− zebrafish compared to WT (n = 3). (**C**) General observation of wild-type and *atm*−/− zebrafish at 6 months old. (**D**) Gender ratio of *atm*−/− compared with WT zebrafish (Over 50 fish each group were calculate). (**E**) HE stains of testis. SG, spermatogonia; SC, spermatocyte; ST, spermatid; SZ, spermatozoa. (**F**) Ratio of each stage of meiosis in *atm*−/− zebrafish testis compared to WT (n = 3). The statistical significance was analyzed using the two-tailed Student’s t-test. Data are shown as means ± SEM.

**Figure 2 biomedicines-10-00392-f002:**
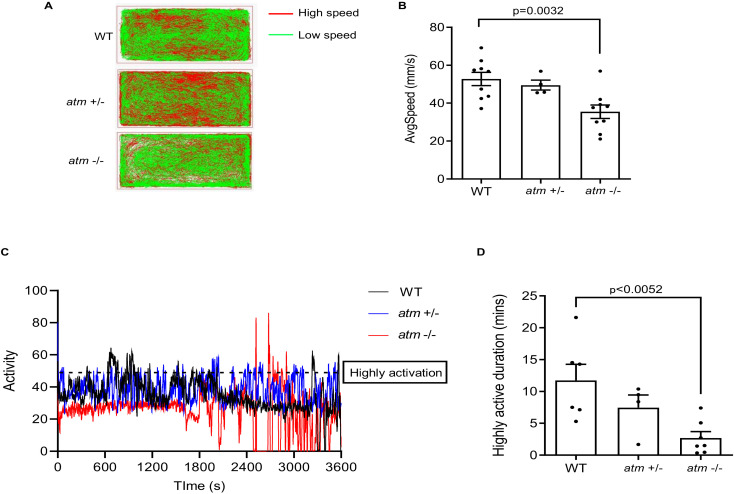
Behavior test of *atm−/−* zebrafish. (**A**) Trajectory diagram of 6-month-old WT, *atm*+/− and *atm*−/− zebrafish in 60-minue open-field tracking test. (**B**) Average speed of 6 months old WT, *atm+/−*, and *atm*−/− zebrafish in **A** (n = 9 for WT and *atm*−/−; n = 4 for *atm+/-*). (**C**) Activity diagram of WT, *atm*+/−, and *atm*−/− zebrafish in 60-minues open-field quantification test. (**D**) Highly activation duration of WT, *atm*+/−, and *atm*−/− zebrafish (n = 6 for WT; n = 7 for *atm*−/−; n = 4 for *atm+/−*). The statistical significance was analyzed using one-way ANOVA. Data are shown as means ± SEM.

**Figure 3 biomedicines-10-00392-f003:**
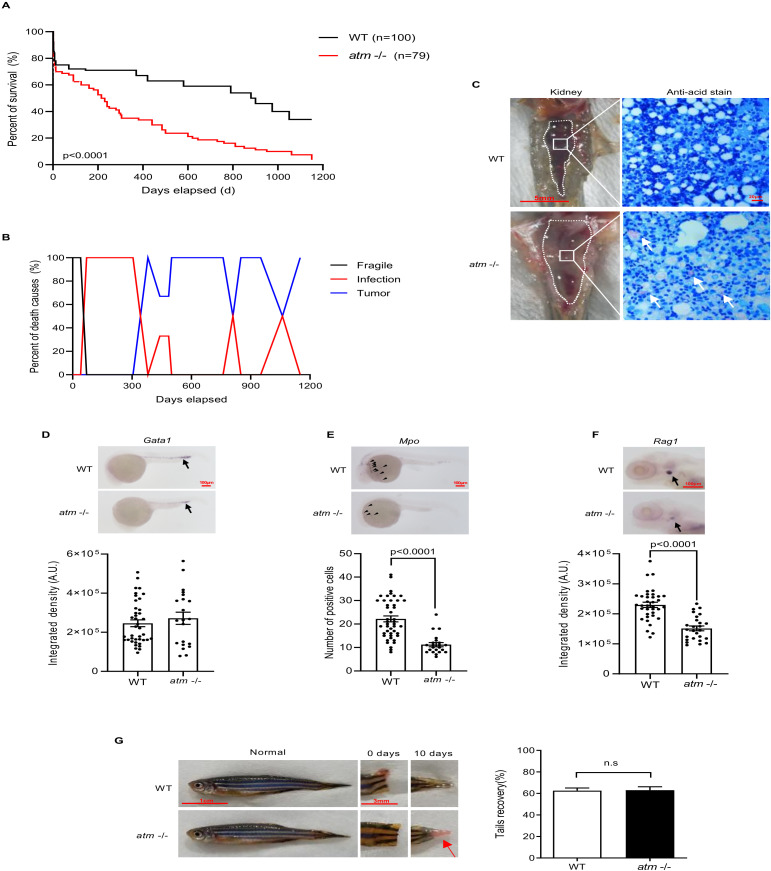
Loss of *atm* in zebrafish causing premature death, immunol deficiency, and regeneration abnormality. (**A**) Life span of WT and *atm*−/− zebrafish. (**B**) Causes of death in WT and *atm*−/− zebrafish. (**C**) Kidney marrow anatomy and anti-acid staining of infective tissue. (**D**–**F**) Whole-embryo in situ hybridization on *atm*−/− and WT zebrafish. (**D**) Representative images of gata1 expression (black arrows) in WT and *atm*−/− embryos at 24-hpf (top). Quantification of the *gata1* signal (bottom) (n = 40 for WT; n = 21 for *atm*−/−). (**E**) Representative images of *mpo* expression (black arrow) in WT and *atm−/−* embryos at 24-hpf (top). Quantification of the *mpo* signals (bottom) (n = 45 for WT; n = 22 for *atm*−/−). (**F**) Representative images of *rag1* expression (black arrow) in WT and *atm*−/− embryos at 5-dpf (top). Quantification of the *rag1* signal (bottom) (n = 36 for WT; n = 26 for *atm*−/−). (**G**) General observation of regeneration ability of *atm*−/− zebrafish’s caudal fin amputation. Ratio of tail recovery is quantified (n = 12 for each group). The statistical significance is analyzed using the two-tailed Student’s t-test. Data are shown in means ± SEM.

**Figure 4 biomedicines-10-00392-f004:**
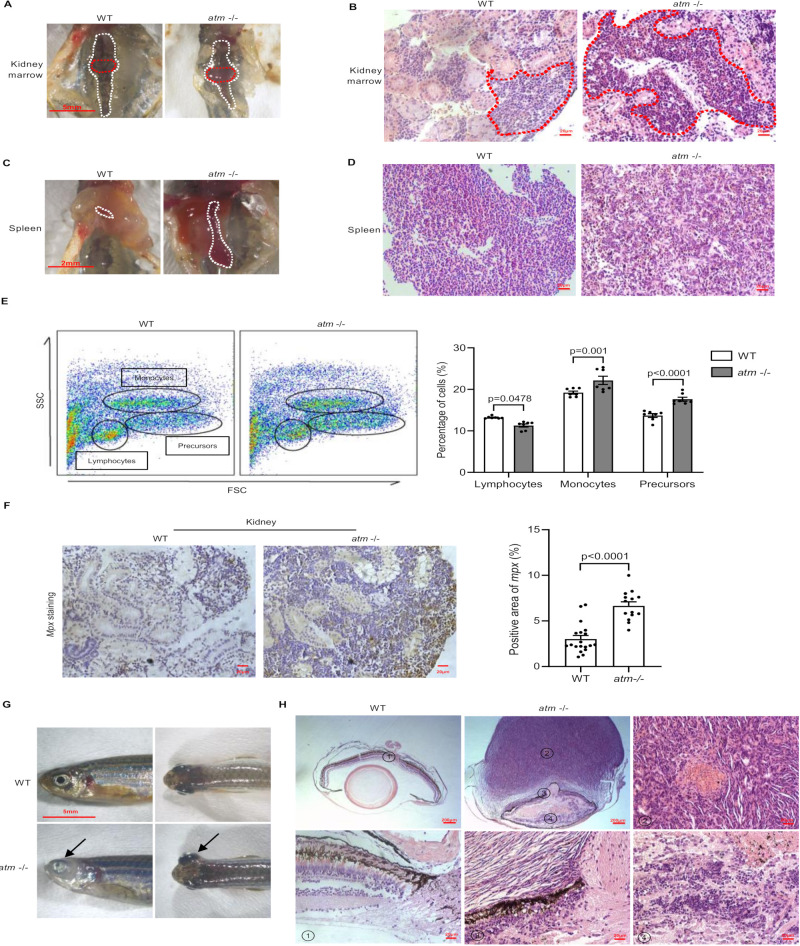
Tumorigenesis in *atm−/−* zebrafish. (**A**) Kidney marrow anatomy of WT and *atm−/−* zebrafish. (**B**) HE staining of kidney marrow shown in A. (**C**) Spleen of WT and *atm−/−* zebrafish. (**D**) HE staining of spleen shown in (**C**). (**E**) Flow cytometry analysis of cells extracted from the kidney of 12-month-old WT and *atm−/−* zebrafish (left). Quantification of percentage of lymphocytes, monocytes, and precursors (right) (n = 7 for each group). (**F**) Immunochemistry of kidney tumor from *atm−/−* fish compared to WT kidney (left). Quantification of mpx-positive signals area (right) (n = 19 for WT; n = 14 for *atm−/−*). (**G**) Neoplasm in the eyes of *atm−/−* zebrafish. (**H**) HE staining of neoplasm in the eyes of *atm−/−* zebrafish compared with the WT eyes. The statistical significance was analyzed using the two-tailed Student’s t-test. Data are shown in means ± SEM.

**Table 1 biomedicines-10-00392-t001:** Statistics of pigment recovery in WT and *atm−/−* fish after amputation.

	Pigment Recovery	
Genotypes	Yes	No
WT	12	0
*atm−/−*	2	10

Fisher’s exact test: *p* value < 0.0001.

## Data Availability

All data and information relevant to this study are available from the corresponding author upon reasonable request.

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
