# Peer review of "Loss of atm in Zebrafish as a Model of Ataxia–Telangiectasia Syndrome"

_biomedicines, 2022, doi:10.3390/biomedicines10020392_

Round 1

Reviewer 1 Report

Manuscript ID: Biomedicines-1510096

Title: Loss of atm in zebrafish as a model of ataxia-telangiectasia syndrome.

The authors engineered a viable zebrafish model with an ataxia-telangiectasia mutated (ATM) gene. Interestingly, atm mutation in zebrafish resulted in different types of tumors than other animal models but much more similar to those in A-T patients, such as malignant peripheral nerve sheath tumors and myeloid leukemia. Like other species, loss of atm affected zebrafish's motor function and coordination. Besides, mutation of atm caused global disorder of immune cell development, impairing the immune capability in adulthood. However, atm-/- zebrafish maintained almost physiological tissue regenerative capacity. The results suggested that immune system impairments in atm homozygote zebrafish are mediated by blockade of hematopoiesis during the development, which may contribute to tumor development and decreased infection defense at adulthood.

The authors developed an atm mutant fish model of A-T syndrome that, in some aspects, seems to be closer to the human A-T disease. The investigation's objectives and results are accurately presented and discussed, so the manuscript is suitable for publication. There are a few clarifications the authors have to address.

  1. Is there any hypothesis that would explain why atm mutation in zebrafish resulted in tumors similar to those seen in A-T patients compared to other animal models?
  2. Regarding ataxia symptoms development, the redox state in the brain and glial cell inflammatory/immune response has been implicated in progressive cerebellar degeneration. Do the authors investigate some of these mechanisms?
  3. Can the authors include a brief paragraph about the role of cellular thiol redox deregulation in ATM oncogenesis?
  4. Please, correct reference 7 (line 456-457).

Reviewer 2 Report

Regarding the manuscript entitled "Loss of atm in zebrafish as a model of ataxia-telangiectasia syndrome", it is very difficult to completely review the document since the referenced figures are missing.

Therefore, a major review is established. However, I make the following comments:

First: The authors must include the figures referred to in the article.

Second: The authors must indicate the reference number and the distributors with the country of location of all the products and devices used in the experiments. To do this, carefully review all sections of the article.

Third: The authors describe some methodology in detail but not others. Therefore, the histochemistry, immunohistochemistry, and flow cytometry sections should be described with special attention.

The authors must describe in detail the marking with Hematoxylin-eosin and Immunohistochemistry, indicating all the steps of the study, as well as the concentration of products, the times in each step, etc.

Authors should indicate the conditions that were used for the observation Flow cytometry.

Round 2

Reviewer 2 Report

Dear authors,

the work has been considerably improved and is proposed for publication in present form.

Best regard